# Spatio-Temporal Distribution of Giant Jellyfish (*Nemopilema nomurai*)

Sunyoung Oh [1], Kyoung-Yeon Kim [2], Hyun-Joo Oh [2], Geunchang Park [1], Wooseok Oh [1] and Kyounghoon Lee [3],*

1   Division of Fisheries Science, Chonnam National University, Yeosu 59696, Korea
2   Oceanic Climate and Ecology Research Division, National Institute of Fisheries Science, Busan 46083, Korea
3   Department of Marine Production Management, Chonnam National University, Yeosu 58754, Korea
*   Correspondence: khlee71@jnu.ac.kr; Tel.: +82-61-659-7124; Fax: +82-61-659-7129

**Abstract:** In this study, the distribution of giant *N*. jellyfish by the echo counting method was investigated in the East China Sea, where giant *N*. jellyfish are known to breed and migrate to the coastal waters of Korea mainly in summer. In addition, the distribution densities by the survey method were compared with the sighting and trawl surveys. In the case of the East China Sea area and the Gijang coastal area, a split beam type scientific echosounder (EK60, Simrad, Norway) and Acoustic data at 38 and 120 kHz were collected while moving at 6–7 kts. In the coastal waters of Korea, acoustics data at 38 and 120 kHz were collected with a split beam type scientific echosounder (EK80, Simrad, Norway) attached to the bottom of the R/V Tamgu No. 21. In the East China Sea, the average distribution densities of acoustic, sighting, and trawl surveys were 8355.7 ($10^{-6}$ ind/m$^3$), 162.2 ($10^{-6}$ ind/m$^3$), and 792.5 ($10^{-6}$ ind/m$^3$), respectively. The average densities in the coastal waters of Korea of acoustic, sighting, and trawl surveys were 2238.7 ($10^{-6}$ ind/m$^3$), 664.9 ($10^{-6}$ ind/m$^3$), and 432.9 ($10^{-6}$ ind/m$^3$), respectively. The average distribution density of the acoustic survey conducted on 21 July 2020 in the coastal waters of Gijang was 1024.5 ($10^{-6}$ ind/m$^3$), and the sighting survey showed 48.8 ($10^{-6}$ ind/m$^3$). The sighting surveys conducted on 22 July 2020 were 393.3 (t) and 19.6 ($10^{-6}$ ind/m$^3$). The average distribution density for the acoustic survey performed on 23 July 2020 was 99.0 ($10^{-6}$ ind/m$^3$), and for the sighting survey was 197.2 ($10^{-6}$ ind/m$^3$). When comparing the results of the acoustic survey with the results of the sighting and the trawl surveys, all surveys except for the survey conducted on 23 July 2020 showed that the acoustic survey was higher than other survey methods.

**Keywords:** acoustic survey; *N*. jellyfish; echo counting method





## 1. Introduction

Giant *N*. jellyfish (*Nemopilema nomurai*) is a cnidarian of the order Jellyfish family, and grows to a height of approximately 1 m or more and a weight of 150 to 200 kg. A rapid increase in sea surface temperature due to global warming is expected in the coastal water of China, as well in the coastal waters of Korea and Japan, causing enormous direct and indirect damage to the marine industry [1,2]. The main reasons for the rapid increase in the number of *N*. jellyfish is due to substantial decreases in the number of the jellyfish's natural enemies, such as turtle and fool fish, and enhanced food competition within the marine ecosystem caused by indiscriminate fishing and environmental pollution.

Recently, various studies have been conducted due to the rapid increase in the population of jellyfish, and a method using sound has been suggested [3]. Methods for estimating the distribution and detecting the presence of jellyfish include trawl, sighting, and acoustic surveys. Although the method of collecting jellyfish by fishing gear and sighting survey are effective in measuring the distribution density of jellyfish, there is a disadvantage that the survey can be carried out only in a specific water layer [4–6].

In order to prevent and reduce the damage caused by jellyfish, it is necessary to grasp various information on the migration path as well as physiological and ecological habits of

jellyfish [7]. The acoustic survey method can identify the distribution of undetectable water layers in a wide area in a short period of time. In addition, considering that zooplankton is the main prey for jellyfish, the distribution in the water layers related to vertical diurnal movement can be identified [5].

Therefore, in this study, the distribution of giant *N.* jellyfish by the echo counting method was investigated in the East China Sea, where giant *N.* jellyfish are known to breed and migrate to the coastal waters of Korea mainly in summer. The distribution densities measured by the sighting and trawl surveys were also compared.

## 2. Materials and Methods

### 2.1. Survey Area and Period

The East China Sea area, which is the migration route of jellyfish, and the coastal waters of Korea, which are introduced in summer, were investigated.

Underwater acoustic survey, sighting survey, and trawl survey were conducted in parallel to understand the distribution of *N.* jellyfish distributed in the East China Sea area and the coastal waters of Korea by the water layer (Figures 1–3). The survey was conducted using Exploration No. 8 (R/V, 282 G/T) from 15 to 28 May, 2020 in the East China Sea, and Exploration No. 21 (R/V, 282 G/T) from July 1 to 16, 2020 in Korea's coastal waters. (R/V, 999 G/T) was used, and the survey was conducted by renting a fishing boat in Gijang from 21 to 23 July, 2020 in the coastal waters of Gijang (Table 1). The collection of acoustic data was carried out at 2–3 knots from the station of the survey, and both sighting and trawl surveys were conducted.

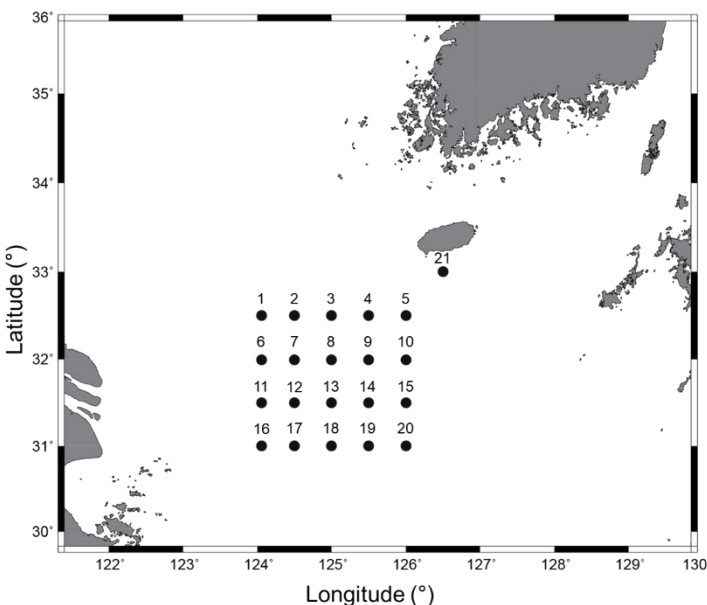

**Figure 1.** Acoustic survey sites in the East China Sea.

**Table 1.** Survey area and period.

| Area | Date | Survey Line (km) | Survey Area (km²) |
|---|---|---|---|
| East China sea | 15–28 May 2020 | 976 | 30,749 |
| Southern sea and Jeju coastal waters | 1–16 July 2020 | 1196 | 24,789 |
| Gijang coastal waters | 21–23 July 2020 | A: 35/B: 28 | A: 26/B: 14 |

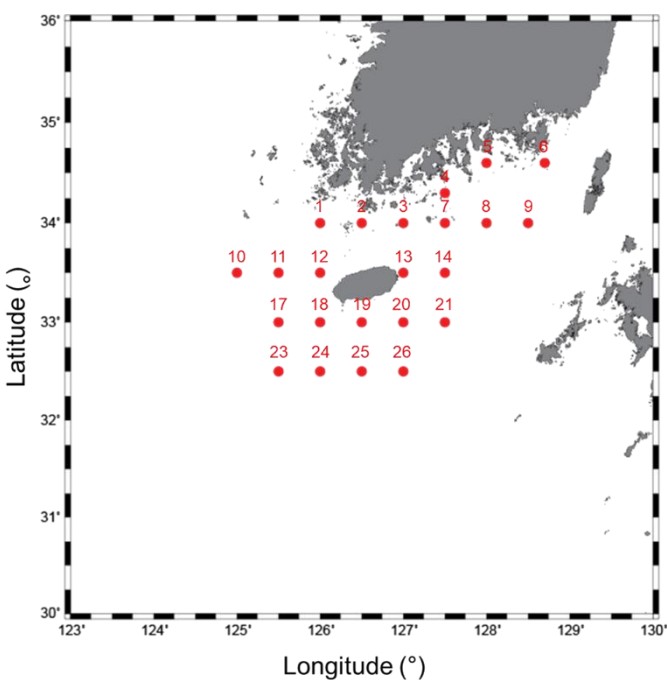

**Figure 2.** Acoustic survey sites in the Southern Sea and Jeju coastal waters.

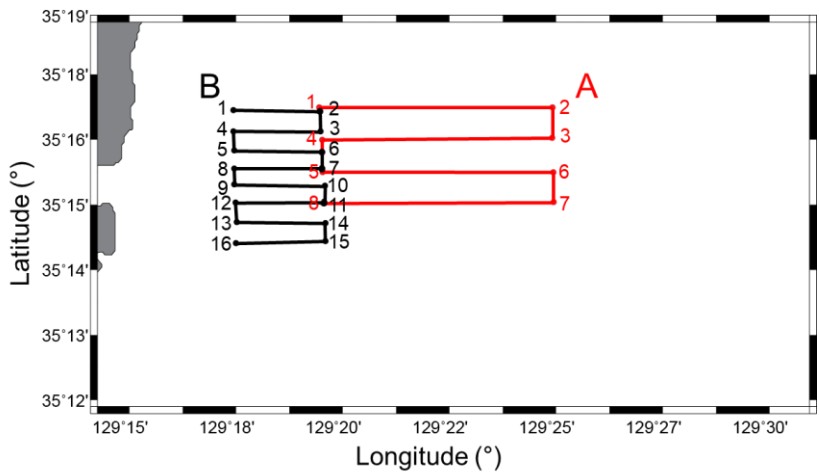

**Figure 3.** Acoustic survey lines in Gijang coastal waters.

## 2.2. Acoustic Equipment System and Data Acquisitions

For the acoustic survey system, in the case of the East China Sea and Gijang coastal waters, a split-beam type scientific echo sounder (EK60, Simrad, Norway) was attached to a tow, and a support stand was used on the prefecture side of the ship to a depth of 1.5 m. The acoustic data of 38 and 120 kHz were collected while moving the ship speed to 6 to 7 kts. In the coastal waters of Korea, sound data at 38 and 120 kHz were collected with a split beam type scientific fish finder (EK80, Simrad, Norway) attached to the bottom of R/V Exploratory Tamgu 21. Acoustic data were collected by setting a pulse width of 1024 ms and a pulse repetition period of 1 s, which are basically recommended in the acoustic resource evaluation. In addition to the acoustic method, the sighting and the trawl surveys were combined to conduct the investigation. For the sighting survey, the number of jellyfish were counted on both the port and starboard sides of the ship at the intervals of 10 m each and a total of 20 m interval. Past atmospheric pressure data were obtained from the Weather Nuri of the Korea Meteorological Administration to investigate the meteorological conditions that facilitates the appearance of *N.* jellyfish.

### 2.3. Acoustic Data Analysis

For the collected acoustic data, the echo signal of jellyfish was extracted after the removal of echogram noise using acoustic analysis post-processing software (Echoview V 8.0, Echoview Software Pty Ltd., Australia) at the laboratory. In general, the density analysis methods to measure the distribution of *N.* jellyfish using sound include the echo integration and echo counting methods. In this study, the echo counting method was used for the density analysis. The echo counting method is usually used when the jellyfish are sparsely distributed and the separate echo signals reflected from each individual jellyfish are received. The separate echo signals appear in the image are then counted to measure the population density of *N.* jellyfish [8].

*N.* jellyfish swims individually and appear as an umbrella-shade-shaped acoustic signal. In this way, the number of jellyfish detected by the scientific echosounder is counted one by one to measure the population, and jellyfish distributed in each water layer can be identified (Figure 4). When identifying the jellyfish signal that appears on the echogram, the TS standard of the jellyfish was used by [9] and counted by citing the study (Figure 5). Since *N.* jellyfish swim individually, the number of jellyfish can be directly counted and distribution in each water layer can be identified.

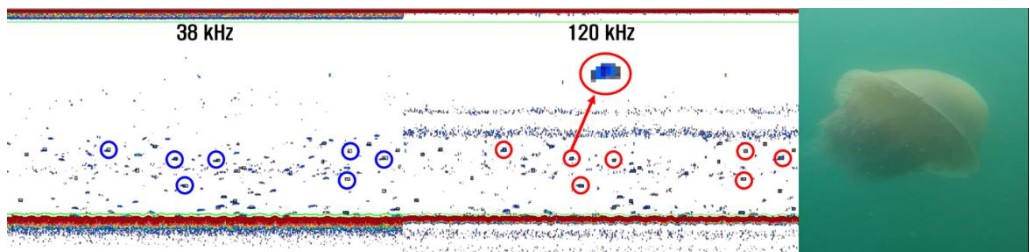

**Figure 4.** Echo signals of jellyfish displayed on the echogram.

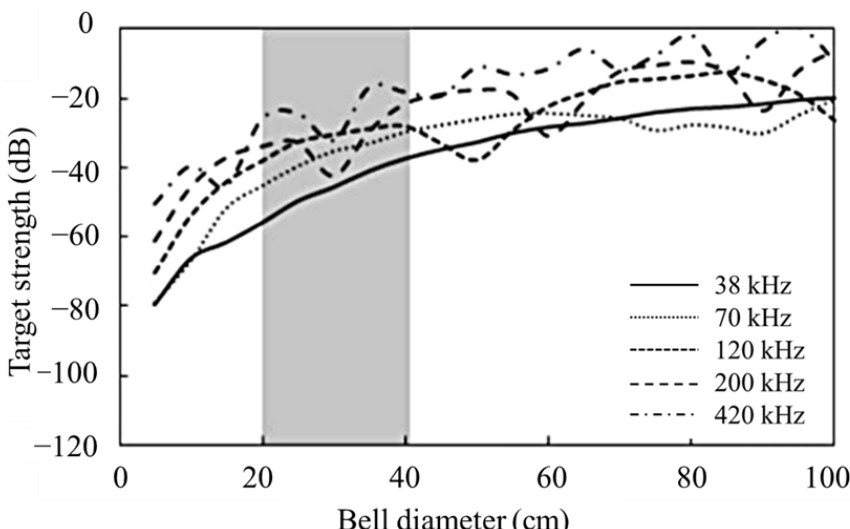

**Figure 5.** Relationship between target strength (TS) and bell diameter of *N.* jellyfish with tilt angle at 38, 70, 120, 200, and 420 kHz [9].

The density of jellyfish detected using echo counting with respect to the survey distance was calculated using Equation (1), and the number of jellyfish detected within the beam width and the survey distance were calculated using Equation (2) by volume [10].

$$\text{Count of Jellyfish's individual} \div \text{Log distance (ind./nmi)} \tag{1}$$

$$\text{Count of Jellyfish's individual} \div \left(\tan 7\deg(°) \times \text{Log distance} \times \text{Avg. depth}\right) \left(\text{ind./m}^3\right) \tag{2}$$

To analyze the density of jellyfish using sighting survey, 10 m sections on both sides of the ship were set, and the transparency of the survey area was averaged. Therefore, it was assumed that jellyfish swimming at a depth of approximately 4 m were detected, and the density was calculated using Equation (3) using the survey distance from the research vessel.

$$\text{Count of Jellyfish}'\text{s individual} \div (\text{Log distance} \times \text{Sighting area}) \left(\text{ind./m}^3\right) \quad (3)$$

The trawl survey was calculated as a population per unit volume to compare the acoustic survey and density. The density of jellyfish using the trawl survey was calculated as shown in Equation (4) using the net width and net height of jellyfish collection net.

$$\text{Count of Jellyfish}'\text{s individual} \div (\text{Log distance} \times \text{Net width} \times \text{Net Height}) \left(\text{ind./m}^3\right) \quad (4)$$

## 3. Results and Discussion

### 3.1. Results of Each Survey Method in the East China Sea

As a result of the acoustic survey, the swimming depth of *N*. jellyfish was found to be in all water layers, and the range of activities was wide (Figure 6). The highest number of *N*. jellyfish was counted as 230 individuals at St. 13. The seawater temperature and salinity ranged between 14.7 and 18.8 °C and between 32.8 and 32.9 psu, respectively. Most jellyfish were distributed in parts of southwest and central.

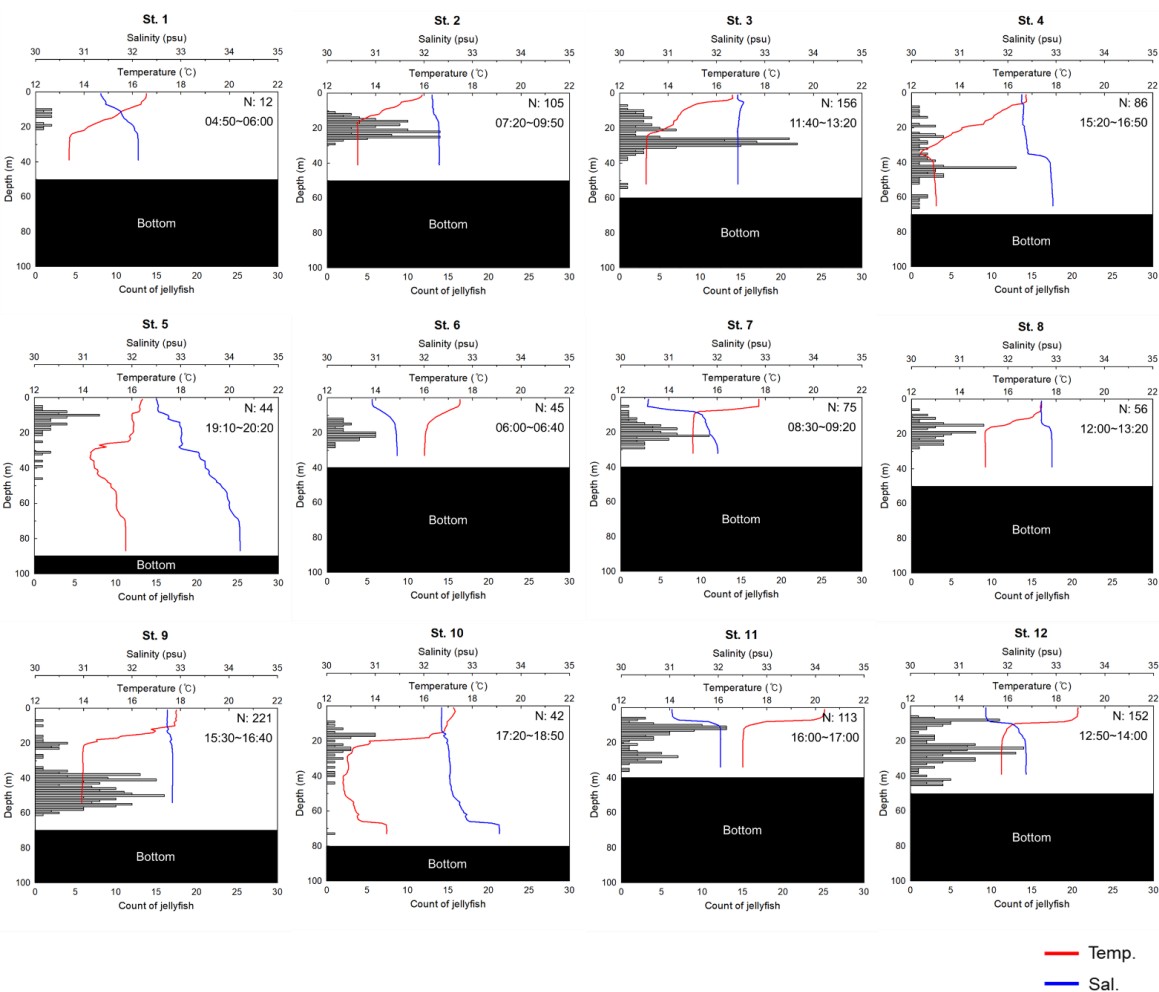

**Figure 6.** *Cont.*

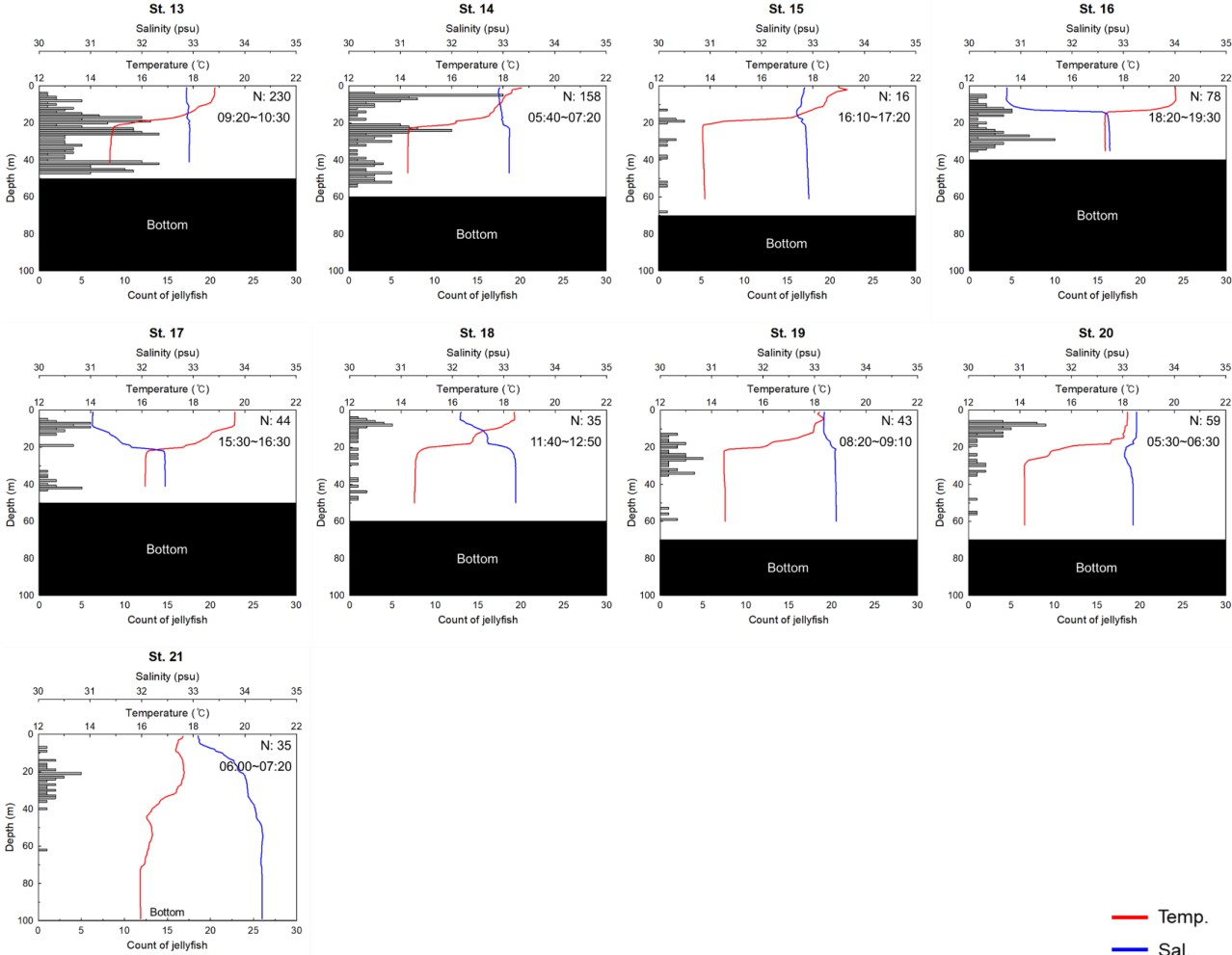

**Figure 6.** Distribution of the water layer in the East China Sea.

As for the density of *N*. jellyfish according to the distance, 158 individuals were detected at St. 13, and the density was 444.0 (ind./nmi). Table 2 shows the distribution density of the jellyfish detected within the scientific echosounder beam width (7 deg.). *N*. Jellyfish exhibited a higher distribution density in the south and west compared to the north, and the highest density was detected in the southeast of St. 14. The average value of the distribution density using the echo counting method was 8355.7 ($10^{-6}$ ind./m$^3$).

As a result of sighting survey, a total of 400 individuals were detected. It appeared most frequently in St. 6, which is near the western part of the survey area (Table 3). When looking at the density according to the irradiation distance, it was 398.0 (ind./nmi) at St. 6 where jellyfish appeared at the highest density. In addition, in order to compare with the results of acoustic irradiation, it is expressed as a distribution density. *N*. jellyfish showed a high distribution density near the west and center of the survey area, and the average value was 162.2 ($10^{-6}$ ind./m$^3$). As a result of comparison with the results of acoustic irradiation, the density of acoustic irradiation was approximately 51 times higher.

The results of the trawl survey were also expressed in terms of distribution density for comparison with the results of the acoustic survey, and the results are shown in Table 4. Similar to acoustic irradiation, it showed high density near the center of the survey area. The average distribution density was 792.5 ($10^{-6}$ ind./m$^3$), which was about 10 times higher for acoustic survey.

**Table 2.** Distribution density of jellyfish using the echo counting method in the East China Sea.

| Site | Ind. | Distance (m/nmi) | Density (ind./nmi) | Density ($10^{-6}$ ind./m$^3$) |
|---|---|---|---|---|
| 1 | 12 | 2013/1.1 | 11.0 | 1517.2 |
| 2 | 105 | 5458/2.9 | 35.6 | 3481.7 |
| 3 | 156 | 3771/2.0 | 76.6 | 5910.8 |
| 4 | 85 | 3192/1.7 | 49.3 | 3098.2 |
| 5 | 44 | 3826/2.1 | 21.3 | 1018.0 |
| 6 | 45 | 1447/0.8 | 57.6 | 6331.9 |
| 7 | 75 | 3213/1.7 | 43.2 | 4752.7 |
| 8 | 56 | 1852/1.0 | 56.0 | 5353.5 |
| 9 | 221 | 1212/0.7 | 337.7 | 23,572.4 |
| 10 | 42 | 4151/2.2 | 18.7 | 1043.1 |
| 11 | 113 | 949/0.5 | 220.5 | 24,244.2 |
| 12 | 152 | 2508/1.4 | 112.2 | 10,730.3 |
| 13 | 230 | 3728/2.0 | 114.3 | 10,690.8 |
| 14 | 158 | 659/0.4 | 444.0 | 36,160.4 |
| 15 | 16 | 4295/2.3 | 6.9 | 439.7 |
| 16 | 78 | 3454/1.9 | 41.8 | 4379.0 |
| 17 | 44 | 756/0.4 | 107.8 | 9673.6 |
| 18 | 35 | 300/0.2 | 216.1 | 16,669.7 |
| 19 | 43 | 1424/0.8 | 55.9 | 3903.6 |
| 20 | 59 | 4548/2.5 | 24.0 | 1488.0 |
| 21 | 35 | 2763/1.5 | 23.5 | 1011.4 |
| Avg. | 1804 | | 98.8 | 8355.7 |

**Table 3.** Distribution density of jellyfish using the sighting survey method in the East China Sea.

| Site | Ind. | Distance (m/nmi) | Density (ind./nmi) | Density ($10^{-6}$ ind./m$^3$) |
|---|---|---|---|---|
| 1 | 35 | 2013/1.1 | 32.2 | 217.3 |
| 2 | 1 | 5458/2.9 | 0.3 | 2.2 |
| 3 | 0 | 3771/2.0 | 0.0 | 0.0 |
| 4 | 0 | 3192/1.7 | 0.0 | 0.0 |
| 5 | 1 | 3826/2.1 | 0.5 | 3.2 |
| 6 | 311 | 1447/0.8 | 398.0 | 2686.5 |
| 7 | 0 | 3213/1.7 | 0.0 | 0.0 |
| 8 | 0 | 1852/1.0 | 0.0 | 0.0 |
| 9 | 11 | 1212/0.7 | 16.8 | 113.4 |
| 10 | 0 | 4151/2.2 | 0.0 | 0.0 |
| 11 | 1 | 949/0.5 | 2.0 | 13.1 |
| 12 | 13 | 2508/1.4 | 9.6 | 64.7 |
| 13 | 7 | 3728/2.0 | 3.5 | 23.4 |
| 14 | 12 | 659/0.4 | 33.7 | 227.6 |
| 15 | 0 | 4295/2.3 | 0.0 | 0.0 |
| 16 | 2 | 3454/1.9 | 1.1 | 7.2 |
| 17 | 1 | 756/0.4 | 2.4 | 16.5 |
| 18 | 0 | 300/0.2 | 0.0 | 0.0 |
| 19 | 2 | 1424/0.8 | 2.6 | 17.5 |
| 20 | 0 | 4548/2.5 | 0.0 | 0.0 |
| 21 | 3 | 2763/1.5 | 2.0 | 13.5 |
| Avg. | 400 | | 24.0 | 162.2 |

**Table 4.** Distribution density of jellyfish using the trawl survey method in the East China Sea.

| Site | Ind. | Density (ind./nmi) | Density ($10^{-6}$ ind./m³) |
|---|---|---|---|
| 2 | 1 | 0.3 | 0.4 |
| 3 | 3 | 1.5 | 10.8 |
| 4 | 0 | 0.0 | 0.0 |
| 5 | 0 | 0.0 | 0.0 |
| 7 | 0 | 0.0 | 0.0 |
| 8 | 2 | 2.0 | 90.7 |
| 9 | 101 | 154.3 | 1174.9 |
| 10 | 0 | 0.0 | 0.0 |
| 11 | 46 | 89.8 | 660.6 |
| 12 | 109 | 80.5 | 1005.9 |
| 13 | 1293 | 642.3 | 5485.1 |
| 14 | 125 | 351.3 | 642.8 |
| 15 | 0 | 0.0 | 0.0 |
| 16 | 0 | 0.0 | 0.0 |
| 17 | 14 | 34.3 | 1463.7 |
| 18 | 135 | 833.4 | 2937.5 |
| 21 | 0 | 0.0 | 0.0 |
| Avg. | 1829 | 128.8 | 792.5 |

*N*. jellyfish were collected using a jellyfish fishing gear in May 2020. The size of the jellyfish was measured after collection, and the size ranged between 3 and 60 cm. The average size of the jellyfish collected was approximately 16.5 cm (Table 5).

**Table 5.** Sampling results of the trawl survey in the East China Sea.

| Site | Ind. | Min. Size | Max. Size | Avg. Size |
|---|---|---|---|---|
| 2 | 1 | 3 | - | 3.0 |
| 3 | 3 | 5 | 22 | 13.7 |
| 8 | 2 | 8 | 12 | 10.0 |
| 9 | 101 | 5 | 43 | 16.3 |
| 11 | 46 | 4 | 32 | 16.6 |
| 12 | 109 | 5 | 46 | 13.8 |
| 13 | 1293 | 4 | 59 | 13.8 |
| 14 | 125 | 7 | 50 | 26.7 |
| 17 | 14 | 6 | 33 | 15.8 |
| 18 | 135 | 5 | 60 | 15.3 |
| total | 1829 | | | 16.5 |

The marine environmental conditions of all survey areas in the East China Sea are shown in Figures 7–9. The seawater temperature and salinity in the major swimming layers of *N*. jellyfish were 12.9–19.5 °C and 31.0–33.9 psu, respectively. The atmospheric pressure when the jellyfish was detected the most was 101,000 Pa, and the atmospheric pressure when the least was detected was 100,500 Pa.

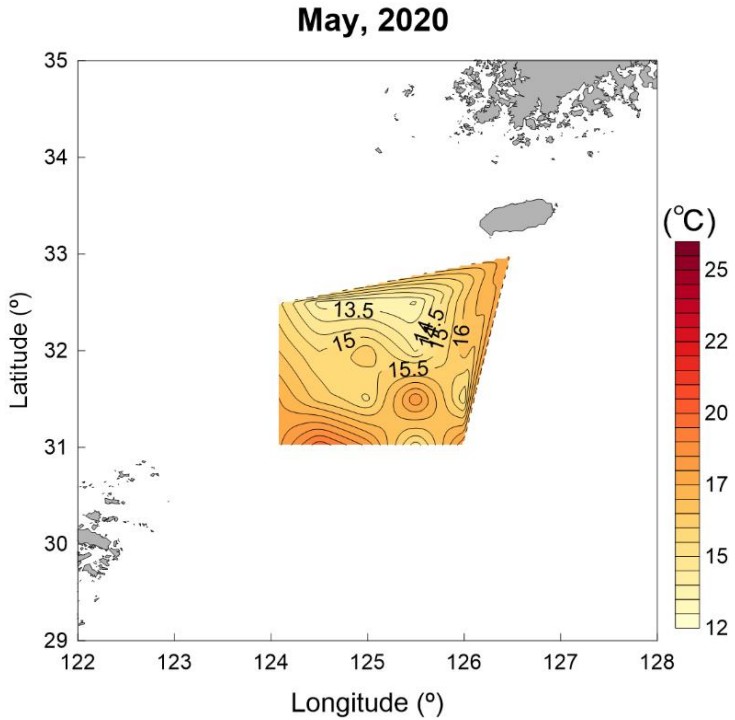

**Figure 7.** Distribution of temperature in the major swimming layers in the East China Sea.

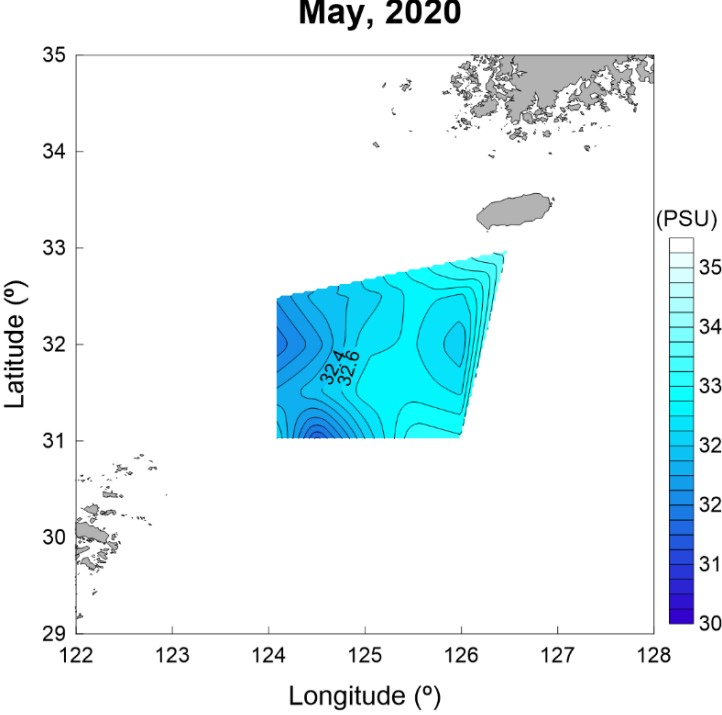

**Figure 8.** Distribution of salinity in the major swimming layers in the East China Sea.

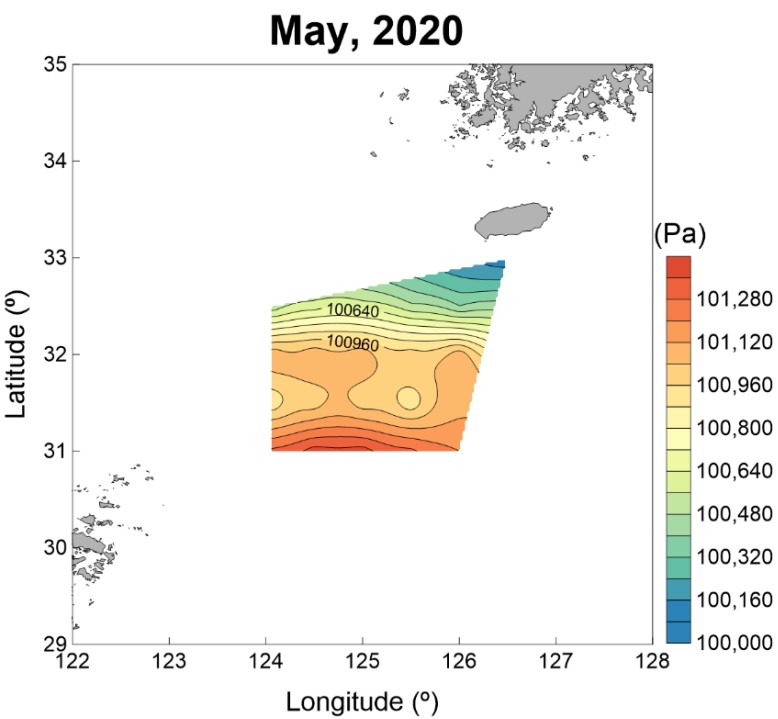

**Figure 9.** Distribution of atmospheric pressure in the East China Sea.

*3.2. Results of Each Survey Method in the Southern Sea and Jeju Coastal Waters*

From 1 to 16 July 2020, acoustic surveys were conducted in the coastal waters of Korea. Figure 10 shows the distribution of *N.* jellyfish in different water layers measured during the maritime survey at the coastal waters of Korea. In July, the equivocal jellyfish signals were revealed using fish detectors due to the influence of weather conditions. Most *N.* jellyfish distributed in July were distributed in all water layers, and a total of 1188 individuals were detected. The highest number of jellyfish individuals was counted as 216 individuals at St. 10, which is located in the west coast of Jeju Island. The seawater temperature and salinity at stations. St. 10 were 12.4–21.1 °C and 30.6–33.1 psu, respectively. *N.* Jellyfish distributed in the coastal waters of Korea were mostly found in the south and west coast of Jeju Island. Table 6 shows the distribution density of *N.* jellyfish distributed in the coastal waters of Korea using the acoustic survey. The density according to the survey distance showed the highest density at 206.0 (ind./nmi) at St. 6 near Geoje. Most of *N.* jellyfish detected within the beam width of the scientific echosounder showed the high densities in the waters around Jeju Island and in the west coast of Korea. The highest population density was found near Geoje Island. The average distribution density of *N.* jellyfish using the echo counting method was 2241.0 ($10^{-6}$ ind./m$^3$).

A total of 4559 *N.* jellyfish individuals were detected in the surface layer during the survey period. In addition, 2049 individuals appeared in St. 11 near Jeju Island. The distribution density measured using the sighting survey is shown in Table 7. Most *N.* jellyfish were distributed in the south coast of Korea and the southwest coastal water of Jeju Island, but the highest distribution density was found in the west coast of Jeju Island, with an average density of 664.9 ($10^{-6}$ ind./m$^3$).

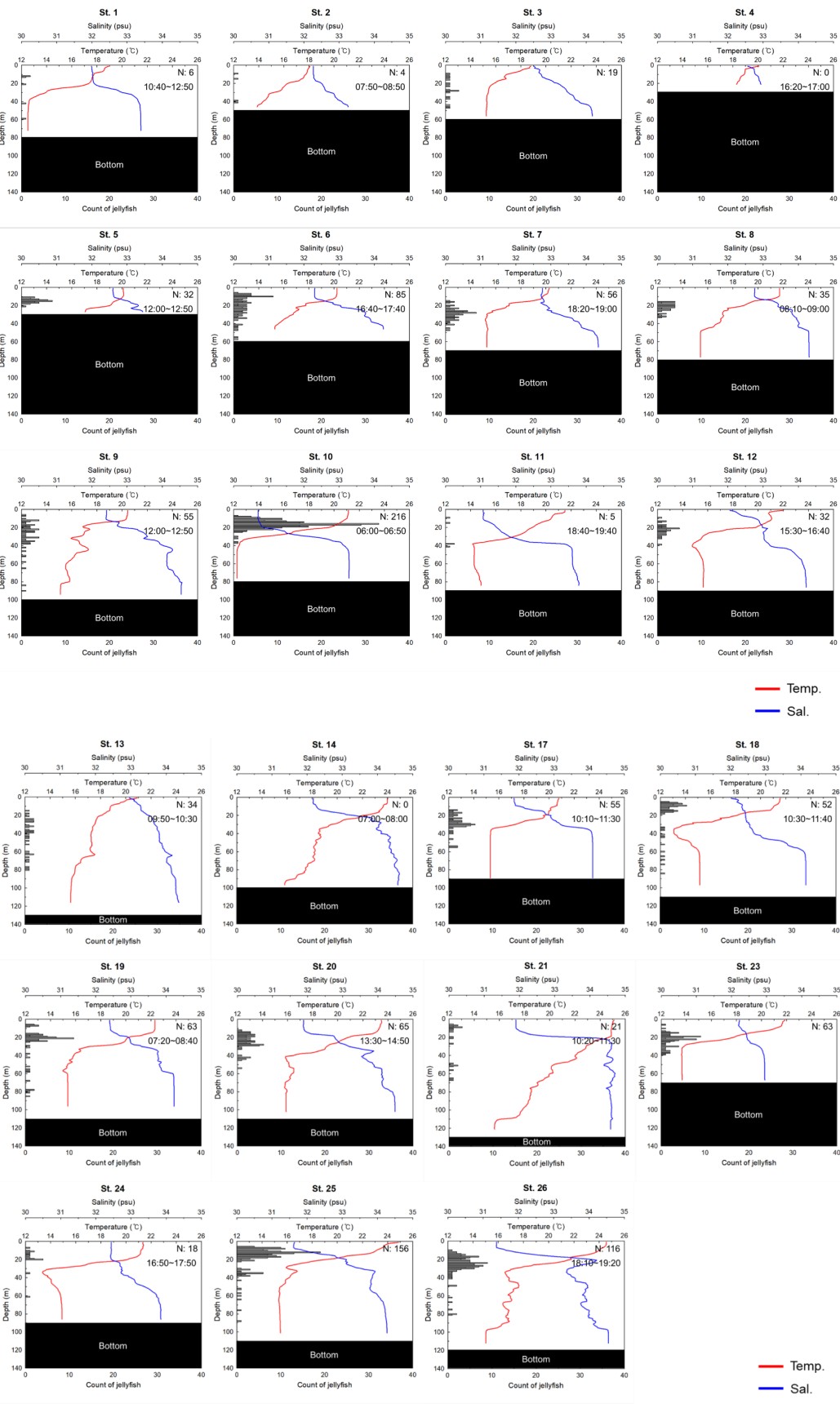

**Figure 10.** Distribution of the water layer in the Southern Sea and Jeju coastal waters.

**Table 6.** Distribution density of jellyfish using the echo counting method in the Southern Sea and Jeju coastal waters.

| Site | Ind. | Distance (m/nmi) | Density (ind./nmi) | Density ($10^{-6}$ ind./m$^3$) |
|---|---|---|---|---|
| 1 | 6 | 5580/3.0 | 2.0 | 112.3 |
| 2 | 4 | 4038/2.2 | 1.8 | 161.4 |
| 3 | 19 | 3941/2.1 | 8.9 | 665.5 |
| 4 | 0 | 3199/1.7 | 0.0 | 0.0 |
| 5 | 32 | 4611/2.5 | 12.9 | 1949.0 |
| 6 | 85 | 764/0.4 | 206.0 | 18,122.2 |
| 7 | 56 | 4722/2.5 | 22.0 | 1509.2 |
| 8 | 35 | 2170/1.2 | 29.9 | 1642.0 |
| 9 | 55 | 2360/1.3 | 43.2 | 2063.1 |
| 10 | 216 | 4098/2.2 | 97.6 | 5503.6 |
| 11 | 5 | 4591/2.5 | 2.0 | 99.7 |
| 12 | 32 | 3181/1.7 | 19.2 | 928.5 |
| 13 | 34 | 2695/1.5 | 23.4 | 796.5 |
| 14 | 0 | 633/0.3 | 0.0 | 0.0 |
| 17 | 55 | 1965/1.1 | 51.8 | 2505.0 |
| 18 | 52 | 2194/1.2 | 43.9 | 1856.0 |
| 19 | 63 | 3903/2.1 | 29.9 | 1276.3 |
| 20 | 65 | 3381/1.8 | 36.2 | 1558.7 |
| 21 | 21 | 1777/1.0 | 21.9 | 723.7 |
| 23 | 63 | 10,646/5.7 | 11.0 | 678.8 |
| 24 | 18 | 624/0.3 | 53.4 | 2581.7 |
| 25 | 156 | 2613/1.4 | 110.6 | 4720.7 |
| 26 | 116 | 3933/2.1 | 54.6 | 2088.8 |
| Avg. | 1188 | | 38.4 | 2241.0 |

**Table 7.** Distribution density of jellyfish using the sighting survey method in the Southern Sea and Jeju coastal waters.

| Site | Ind. | Distance (m/nmi) | Density (ind./nmi) | Density ($10^{-6}$ ind./m$^3$) |
|---|---|---|---|---|
| 1 | 257 | 5580/3.0 | 85.3 | 575.7 |
| 2 | 157 | 4038/2.2 | 72.0 | 486.0 |
| 3 | 239 | 3941/2.1 | 112.3 | 758.0 |
| 4 | 0 | 3199/1.7 | 0.0 | 0.0 |
| 5 | 187 | 4611/2.5 | 75.1 | 506.9 |
| 6 | 28 | 764/0.4 | 67.9 | 458.1 |
| 7 | 130 | 4722/2.5 | 51.0 | 344.1 |
| 8 | 1 | 2170/1.2 | 0.9 | 5.7 |
| 9 | 33 | 2360/1.3 | 25.9 | 174.7 |
| 10 | 137 | 4098/2.2 | 61.9 | 417.8 |
| 11 | 2049 | 4591/2.5 | 826.6 | 5578.8 |
| 12 | 1153 | 3181/1.7 | 671.3 | 4530.8 |
| 13 | 9 | 2695/1.5 | 6.2 | 41.7 |
| 14 | 0 | 633/0.3 | 0.0 | 0.0 |
| 17 | 61 | 1965/1.1 | 57.5 | 388.0 |
| 18 | 2 | 2194/1.2 | 1.7 | 11.3 |
| 19 | 5 | 3903/2.1 | 2.4 | 16.0 |
| 20 | 3 | 3381/1.8 | 1.6 | 11.0 |
| 21 | 0 | 1777/1.0 | 0.0 | 0.00 |
| 23 | 51 | 10,646/5.7 | 8.9 | 59.8 |
| 24 | 43 | 624/0.3 | 127.6 | 861.3 |
| 25 | 14 | 2613/1.4 | 9.9 | 66.9 |
| 26 | 0 | 3933/2.1 | 0.0 | 0.0 |
| Avg. | 4559 | | 98.5 | 664.9 |

Table 8 shows the results of the distribution densities of *N.* jellyfish using the trawl survey. Most of the jellyfish showed the high distribution density in the south coast of Korea and the west coast of Jeju Island. In particular, the jellyfish showed the highest density near Wando and Goheung, and the average distribution density was 432.9 ($10^{-6}$ ind./m$^3$).

**Table 8.** Distribution density of jellyfish using the trawl survey method in the Southern Sea and Jeju coastal waters.

| Site | Ind. | Density (ind./nmi) | Density ($10^{-6}$ ind./m$^3$) |
|---|---|---|---|
| 1 | 203 | 67.4 | 612.1 |
| 3 | 152 | 71.4 | 2036.1 |
| 5 | 22 | 8.8 | 265.2 |
| 6 | 85 | 206.0 | 584.3 |
| 8 | 0 | 0.0 | 0.0 |
| 9 | 44 | 34.5 | 579.4 |
| 10 | 16 | 7.2 | 154.8 |
| 11 | 131 | 52.8 | 1383.1 |
| 12 | 97 | 56.5 | 1409.5 |
| 14 | 0 | 0.0 | 0.0 |
| 17 | 15 | 14.1 | 231.3 |
| 18 | 29 | 24.5 | 342.9 |
| 19 | 2 | 0.9 | 31.6 |
| 20 | 1 | 0.5 | 12.7 |
| 21 | 0 | 0.0 | 0.0 |
| 23 | 65 | 11.3 | 472.2 |
| 24 | 3 | 8.9 | 85.7 |
| 25 | 2 | 1.4 | 24.5 |
| 26 | 0 | 0.0 | 0.0 |
| Avg. | 867 | 29.8 | 432.9 |

Table 9 shows the results of the size measured for the jellyfish collected during the trawl survey. The size of the jellyfish collected in July 2020 ranged between 25 and 100 cm, with the average of 56.5 cm.

**Table 9.** Sampling results of the trawl survey in the Southern Sea and Jeju coastal waters.

| Site | Ind. | Min. Size | Max. Size | Avg. Size |
|---|---|---|---|---|
| 1 | 50 | 30 | 70 | 45.0 |
| 3 | 55 | 27 | 90 | 57.4 |
| 5 | 17 | 30 | 85 | 54.6 |
| 6 | 27 | 25 | 97 | 57.3 |
| 9 | 28 | 38 | 100 | 60.6 |
| 10 | 13 | 48 | 99 | 67.6 |
| 11 | 44 | 33 | 67 | 50.3 |
| 12 | 50 | 26 | 70 | 46.3 |
| 17 | 12 | 36 | 73 | 59.3 |
| 18 | 23 | 27 | 73 | 52.2 |
| 19 | 2 | 55 | 81 | 68.0 |
| 20 | 1 | 32 | - | 32.0 |
| 23 | 38 | 37 | 95 | 70.2 |
| 24 | 3 | 60 | 65 | 63.3 |
| 25 | 2 | 56 | 73 | 64.5 |
| total | 365 | | | 56.5 |

The marine environmental conditions of the coastal waters of Korea is shown in Figures 11–13. Seawater temperature and salinity were distributed at 14.1–24.9 °C and

30.7–34.0 psu, respectively. Atmospheric pressure at St. 10, where jellyfish were detected the most, was 100,400 Pa.

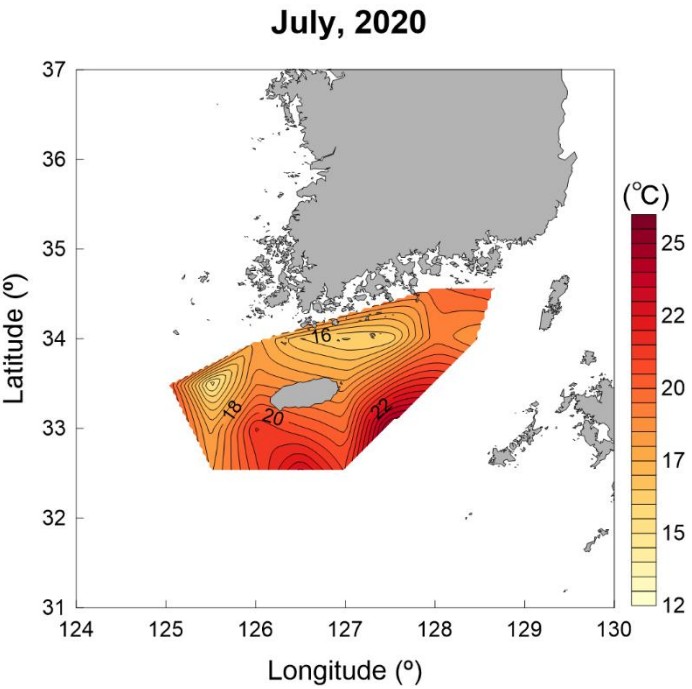

**Figure 11.** Distribution of temperature in the major swimming layers in the Southern Sea and Jeju coastal waters.

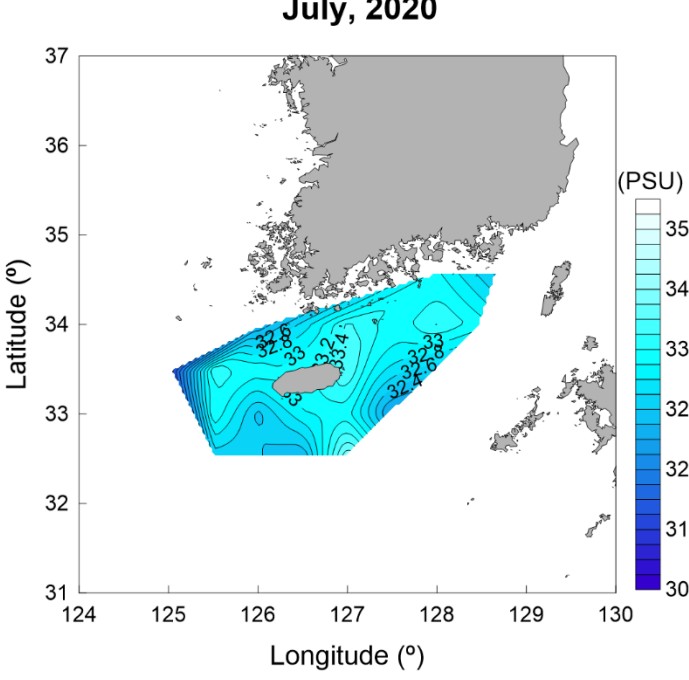

**Figure 12.** Distribution of salinity in the major swimming layers in the Southern Sea and Jeju coastal waters.

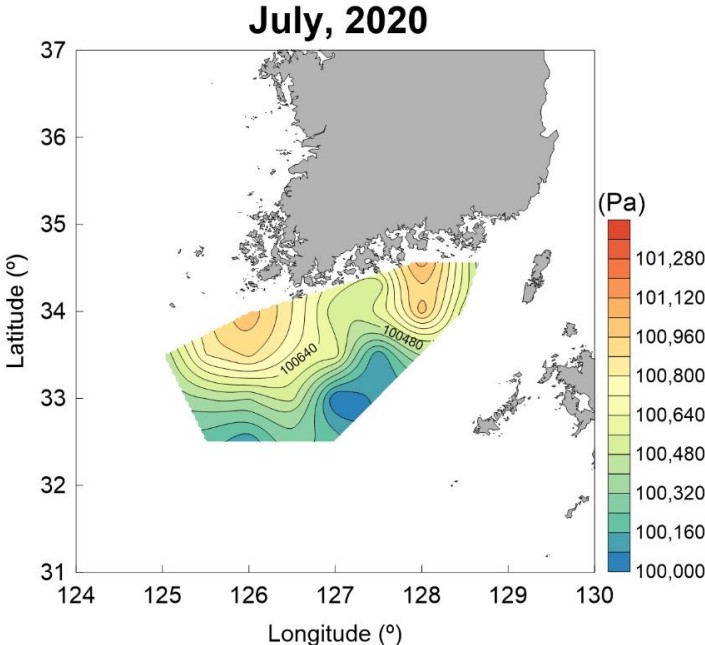

**Figure 13.** Distribution of atmospheric pressure in the Southern Sea and Jeju coastal waters.

The size of the jellyfish collected in July 2020 was approximately three times larger than the jellyfish collected in May 2020. The range of inhabiting seawater temperature and salinity became wider.

### 3.3. Results of Each Survey Method in the Coastal Waters of Gijang

Figure 14 shows the results of the distribution of the acoustic survey conducted in the coastal waters of Gijang between 21 July 2020.

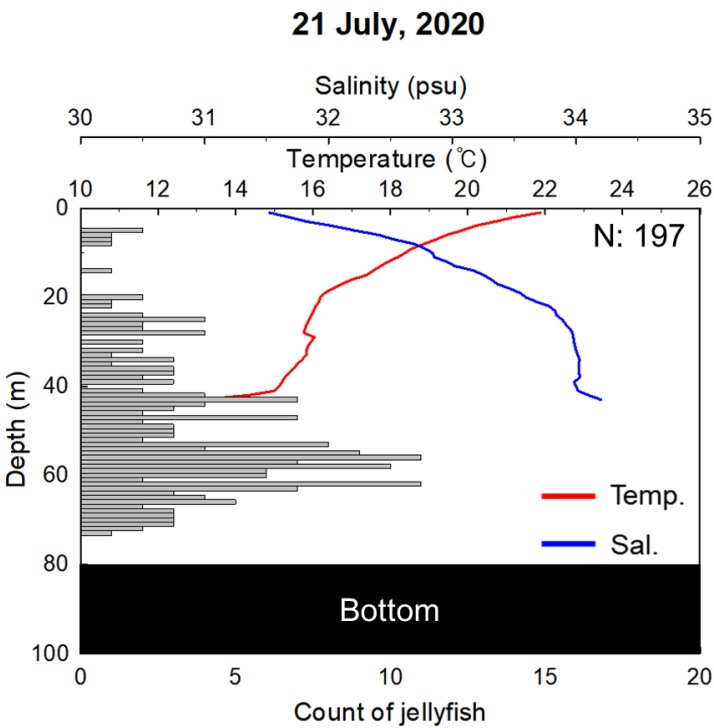

**Figure 14.** Distribution by water layers in the Gijang coastal waters sea (21 July 2020).

The results of the acoustic survey conducted on 21 July 2020 showed that the jellyfish were distributed throughout the entire water layer, and a total of 197 individuals were detected. Most of the giant jellyfish were found below 40 m depth. Table 10 shows the distribution density of the jellyfish using the acoustic survey. The population density of the jellyfish was at the highest in the northern part of the surveyed area, and relatively low in the southern part. The average value of the distribution density measured using the acoustic survey was 1024.5 ($10^{-6}$ ind/m$^3$).

**Table 10.** Distribution density of jellyfish using the echo counting method in the Gijang coastal waters sea (7/21).

| Site | Ind. | Distance (m/nmi) | Density (ind./nmi) | Density ($10^{-6}$ ind./m$^3$) |
|---|---|---|---|---|
| Line 1 | 73 | 8008/4.3 | 16.9 | 1726.5 |
| Line 2 | 49 | 7889/4.3 | 11.5 | 936.7 |
| Line 3 | 64 | 7933/4.3 | 14.9 | 1263.5 |
| Line 4 | 11 | 7920/4.3 | 2.6 | 171.3 |
| Avg. | 197 | | 11.5 | 1024.5 |

Table 11 shows the results of distribution density of the jellyfish using the sighting survey. As a result, the density was higher in the south than in the north, and the average distribution density was 48.8 ($10^{-6}$ ind/m$^3$). In addition, a total of 124 individuals were counted in the surface layer.

**Table 11.** Distribution density of jellyfish using the sighting survey method in the Gijang coastal waters sea (7/21).

| Site | Ind. | Distance (m/nmi) | Density (ind./nmi) | Density ($10^{-6}$ ind./m$^3$) |
|---|---|---|---|---|
| Line 1 | 14 | 8008/4.3 | 3.2 | 21.8 |
| Line 2 | 14 | 7889/4.3 | 3.3 | 22.1 |
| Line 3 | 35 | 7933/4.3 | 8.2 | 55.1 |
| Line 4 | 61 | 7920/4.3 | 14.3 | 96.2 |
| Avg. | 124 | | 7.2 | 48.8 |

The results of the distribution density of the acoustic and sighting surveys were compared. The distribution density measured using the acoustic survey showed a higher density in the northern coastal waters compared to the southern areas of the surveyed area. Whereas the sighting survey results showed a higher density in the southern coastal waters of the surveyed area as opposed to the acoustic survey results.

The marine environmental conditions of all survey stations conducted on July 21 is shown in Figures 15–17. The seawater temperature and salinity, where most N. jellyfish were detected, were 13.1–16.1 °C and 33.9–34.3 psu, respectively, and the average atmospheric pressure was 101,100 Pa.

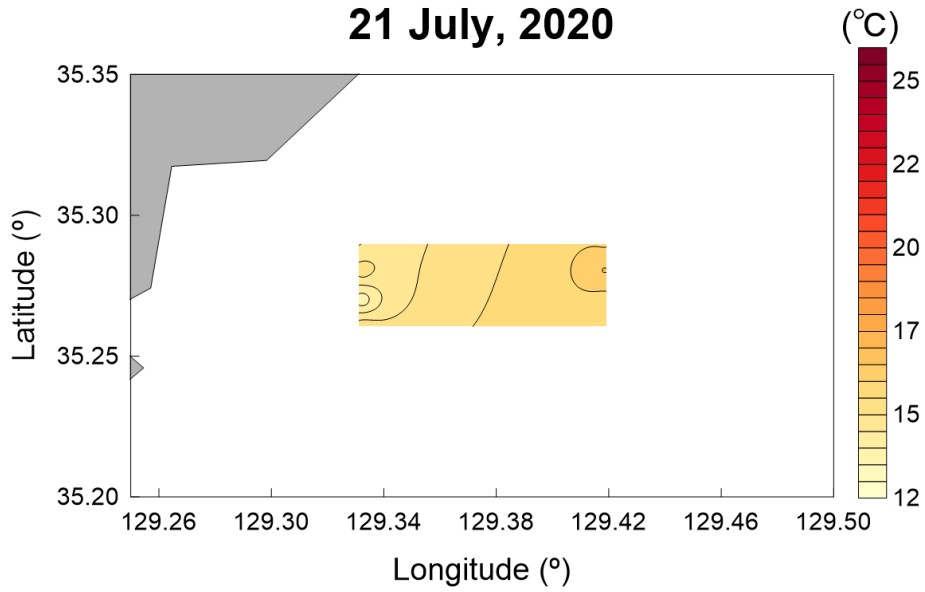

**Figure 15.** Distribution of temperature in the major swimming layers in the Gijang coastal waters sea (21 July 2020).

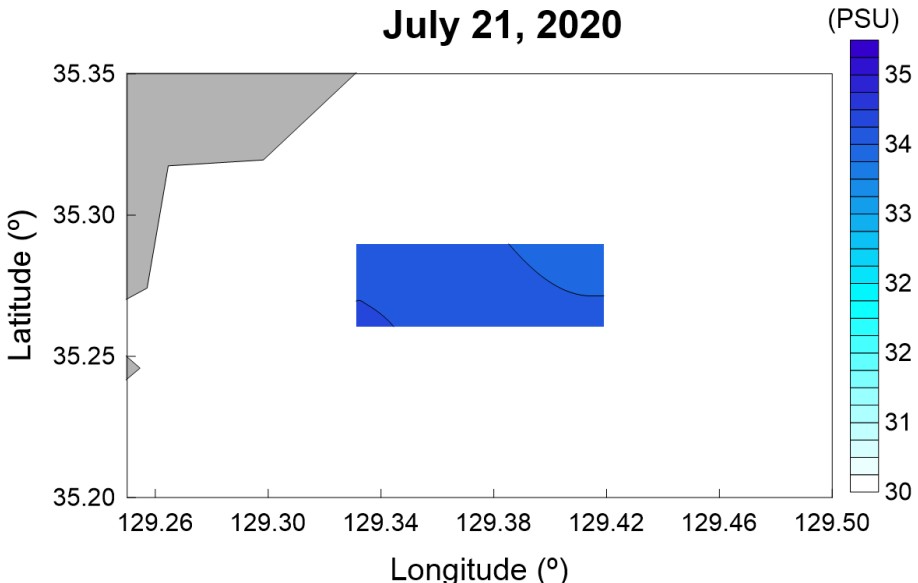

**Figure 16.** Distribution of salinity in the major swimming layers in the Gijang coastal waters sea (21 July 2020).

Figure 18 shows the results of the distribution of the acoustic survey conducted in the coastal waters of Gijang between 22 July 2020.

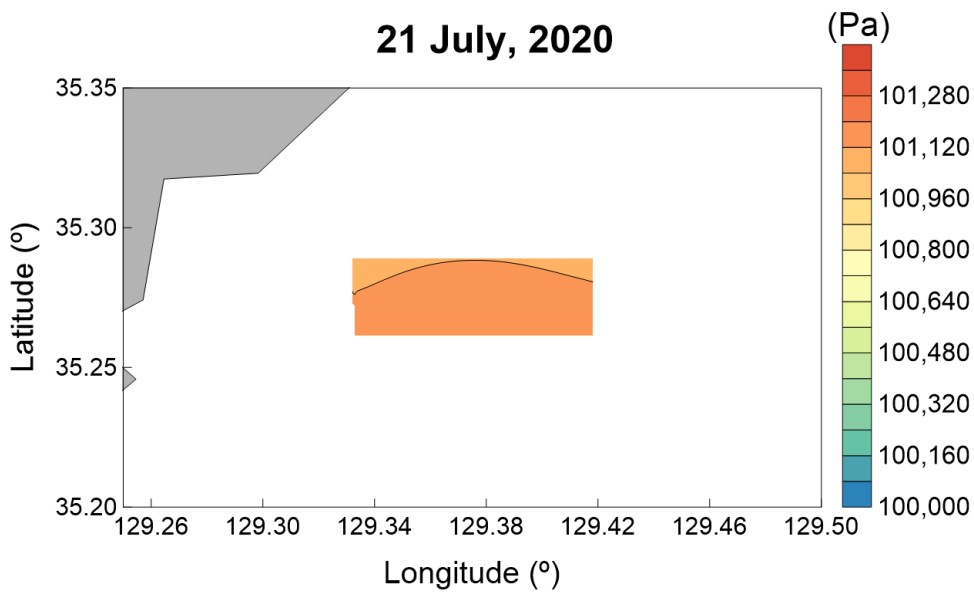

**Figure 17.** Distribution of atmospheric pressure in the Gijang coastal waters sea (21 July 2020).

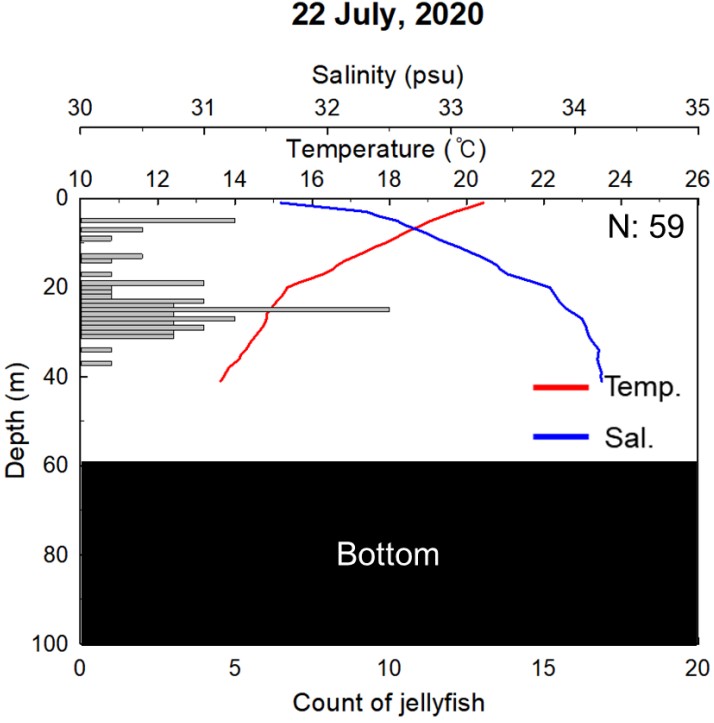

**Figure 18.** Distribution by water layers in the Gijang coastal waters sea (22 July 2020).

As a result of the acoustic survey on 22 July 2020, a total of 59 individuals were detected by the scientific echosounder, and they were distributed in the vicinity of 5–40 m. The distribution density of the jellyfish detected in the coastal waters of Gijang was not detected near the south but showed a high distribution density near the north. The average distribution density of the acoustic irradiation method was 393.3 ($10^{-6}$ ind/m³) (Table 12).

A total of 38 individuals were counted via the sighting survey, which did not appear much in the surface layer. In contrast to the results of the acoustic survey, the distribution density of jellyfish was more distributed in the southern area than in the northern part of the surveyed area (Table 13). The average distribution density was 19.6 ($10^{-6}$ ind/m³).

**Table 12.** Distribution density of jellyfish using the echo counting method in the Gijang coastal waters sea (7/22).

| Site | Ind. | Distance (m/nmi) | Density (ind./nmi) | Density ($10^{-6}$ ind./m$^3$) |
|---|---|---|---|---|
| Line 1 | 32 | 2954/2 | 20.1 | 1764.5 |
| Line 2 | 7 | 2967/2 | 4.4 | 408.8 |
| Line 3 | 11 | 2982/2 | 6.8 | 546.2 |
| Line 4 | 6 | 2980/2 | 3.7 | 303.6 |
| Line 5 | 1 | 3087/2 | 0.6 | 43.9 |
| Line 6 | 0 | 3013/2 | 0.0 | 0.0 |
| Line 7 | 0 | 3034/2 | 0.0 | 0.0 |
| Line 8 | 2 | 3062/2 | 1.2 | 79.4 |
| Avg. | 59 | | 4.6 | 393.3 |

**Table 13.** Distribution density of jellyfish using the sighting method in the Gijang coastal waters sea (7/22).

| Site | Ind. | Distance (m/nmi) | Density (ind./nmi) | Density ($10^{-6}$ ind./m$^3$) |
|---|---|---|---|---|
| Line 1 | 0 | 2954/2 | 0.0 | 0.0 |
| Line 2 | 2 | 2967/2 | 1.2 | 8.4 |
| Line 3 | 2 | 2982/2 | 1.2 | 8.3 |
| Line 4 | 5 | 2980/2 | 3.1 | 20.9 |
| Line 5 | 0 | 3087/2 | 0.0 | 0.0 |
| Line 6 | 9 | 3013/2 | 5.5 | 37.3 |
| Line 7 | 1 | 3034/2 | 0.6 | 4.1 |
| Line 8 | 19 | 3062/2 | 11.5 | 77.5 |
| Avg. | 38 | | 3.9 | 19.6 |

The ranges of seawater temperature and salinity, where most of the jellyfish were detected on 22 July 2020, were 13.3–18.9 °C and 32.6–34.2 psu, respectively, and the average atmospheric pressure was 100,900–101,200 Pa. (Figures 19–21).

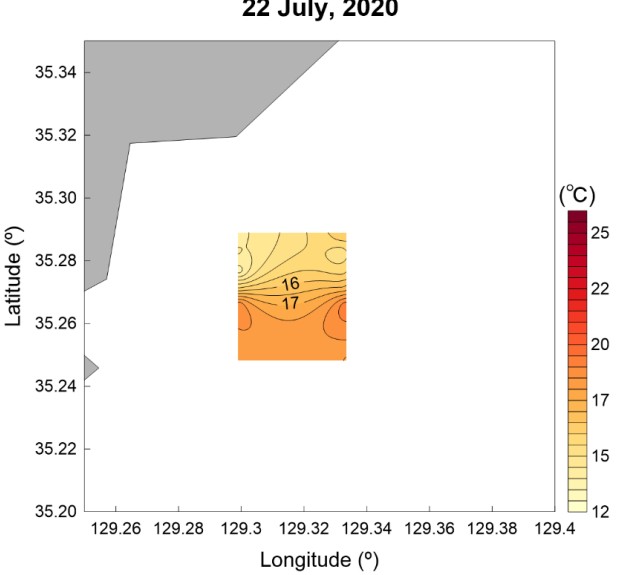

**Figure 19.** Distribution of temperature in the major swimming layers in the Gijang coastal waters sea (22 July 2020).

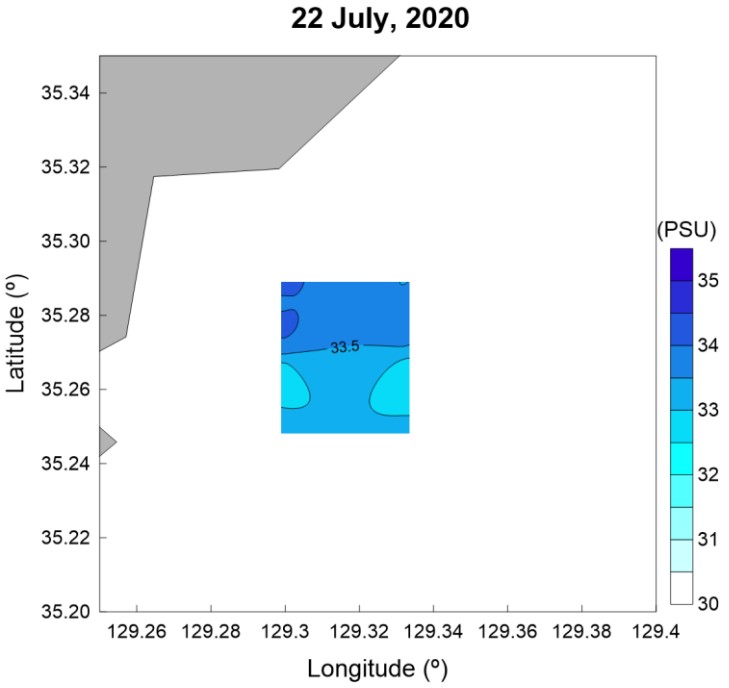

**Figure 20.** Distribution of salinity in the major swimming layers in the Gijang coastal waters sea (22 July 2020).

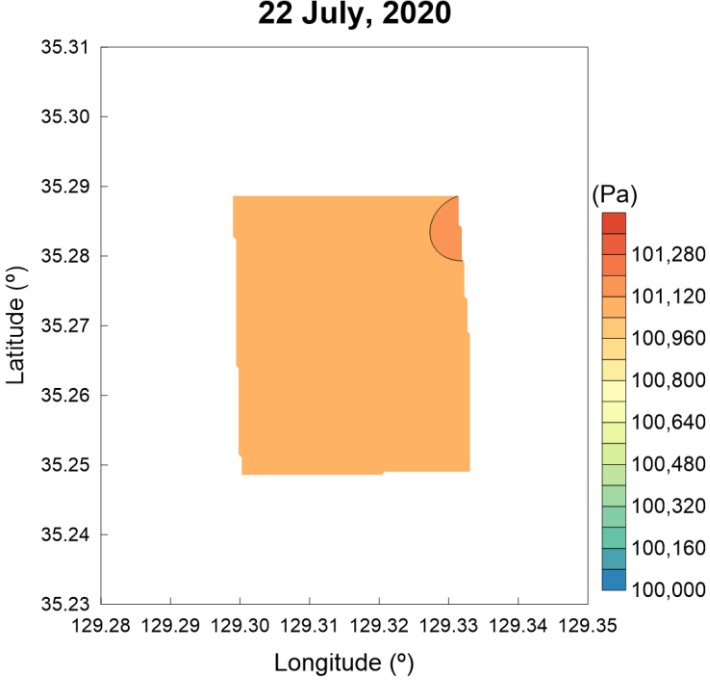

**Figure 21.** Distribution of atmospheric pressure in the Gijang coastal waters sea (22 July 2020).

Figure 22 shows the results of the distribution of the acoustic survey conducted in the coastal waters of Gijang between 23 July 2020.

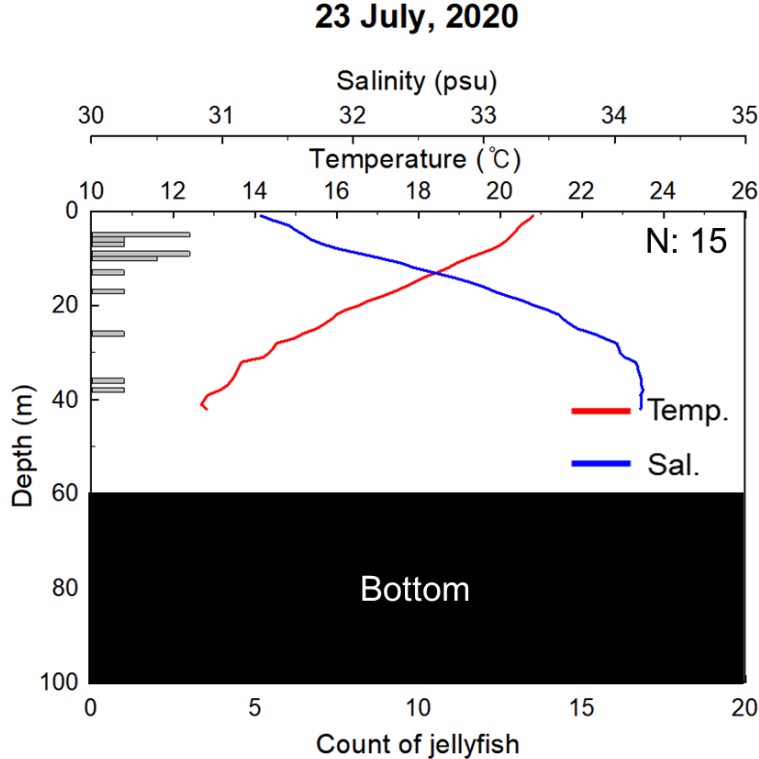

**Figure 22.** Distribution by water layers in the Gijang coastal waters sea (23 July 2020).

The acoustic survey conducted on 23 July 2020 was conducted in the morning like the acoustic survey conducted on the 22 July 2020, and a total of 15 individuals were detected. In general, N. jellyfish diurnally migrate between 5 and 30 m depth. Table 14 shows the results of the distribution density using the acoustic survey. N. jellyfish showed a high density at the northern and southern ends of the survey area, and low near the center. The average distribution density of the acoustic survey results was 99.0 ($10^{-6}$ ind/m$^3$).

**Table 14.** Distribution density of jellyfish using the echo counting method in the Gijang coastal waters sea (7/23).

| Site | Ind. | Distance (m/nmi) | Density (ind./nmi) | Density ($10^{-6}$ ind./m$^3$) |
|---|---|---|---|---|
| Line 1 | 6 | 2954/2 | 3.8 | 352.0 |
| Line 2 | 1 | 2967/2 | 0.6 | 58.4 |
| Line 3 | 0 | 2982/2 | 0.0 | 0.0 |
| Line 4 | 0 | 2980/2 | 0.0 | 0.0 |
| Line 5 | 0 | 3087/2 | 0.0 | 0.0 |
| Line 6 | 2 | 3013/2 | 1.2 | 91.6 |
| Line 7 | 0 | 3034/2 | 0.0 | 0.0 |
| Line 8 | 6 | 3062/2 | 3.6 | 290.2 |
| Avg. | 15 | | 1.2 | 99.0 |

In the sighting survey, 384 individuals were detected on the surface layer. The distribution density of the sighting survey performed on July 23 is shown in Table 15. In general, the distribution density was mostly even. It was relatively low in the northern seas, and the highest density was found near the center. The average distribution density was 197.2 ($10^{-6}$ ind/m$^3$).

**Table 15.** Distribution density of jellyfish using the sighting method in the Gijang coastal waters sea (7/23).

| Site | Ind. | Distance (m/nmi) | Density (ind./nmi) | Density ($10^{-6}$ ind./m$^3$) |
|------|------|------------------|--------------------|-------------------------------|
| Line 1 | 0 | 2954/2 | 0.0 | 0.0 |
| Line 2 | 0 | 2967/2 | 0.0 | 0.0 |
| Line 3 | 16 | 2982/2 | 9.9 | 67.0 |
| Line 4 | 67 | 2980/2 | 41.6 | 281.0 |
| Line 5 | 123 | 3087/2 | 73.8 | 498.0 |
| Line 6 | 30 | 3013/2 | 18.4 | 124.4 |
| Line 7 | 85 | 3034/2 | 51.9 | 350.2 |
| Line 8 | 63 | 3062/2 | 38.1 | 257.1 |
| Avg. | 384 | | 39.0 | 197.2 |

The marine environmental conditions of the main swimming depths of the jellyfish found on 23 July 2020 are shown in Figures 23–25. The seawater temperature of the main swimming layer ranged between 19.1–21.1 °C, the salinity ranged between 31.3–32.2 psu. The atmospheric pressure ranged between 100,900–101,200 Pa.

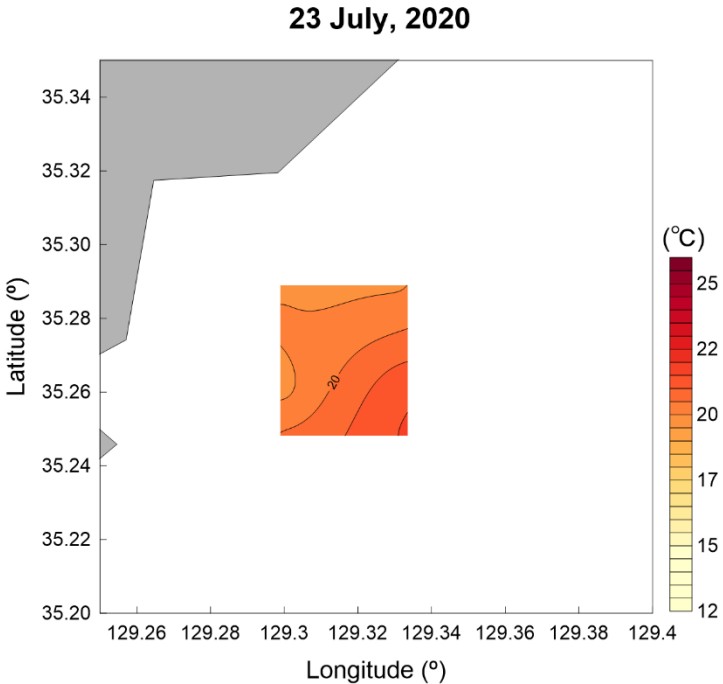

**Figure 23.** Distribution of temperature in the major swimming layers in the Gijang coastal waters sea (23 July 2020).

On 21 July 2020, the average atmospheric pressure was 101,100 Pa and the average precipitation was 3.4 mm. The average atmospheric pressure was 101,000–101,100 Pa and the average precipitation was 105.3 mm on 22 July 2020. On 23 July 2020, the average atmospheric pressure was 100,900–101,200 Pa and the average precipitation was 176.2 mm. Considering the sunnier weather condition with little precipitation on 21 July 2020 compared to 22 and 23 July 2020, the higher number of jellyfish were detected using the acoustic survey. In contrast, the sighting survey resulted in a higher number of individuals being detected in the surface layer during rainy days.

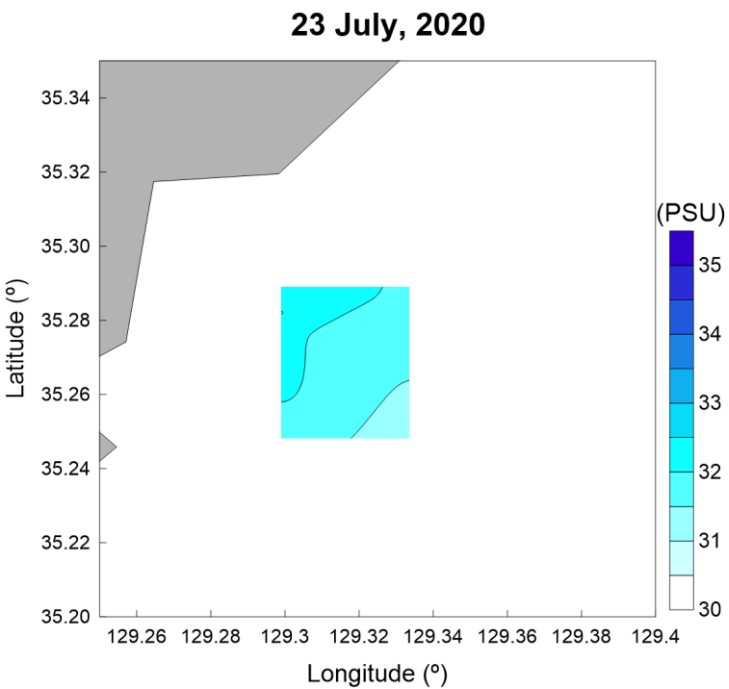

**Figure 24.** Distribution of salinity in the major swimming layers in the Gijang coastal waters sea (23 July 2020).

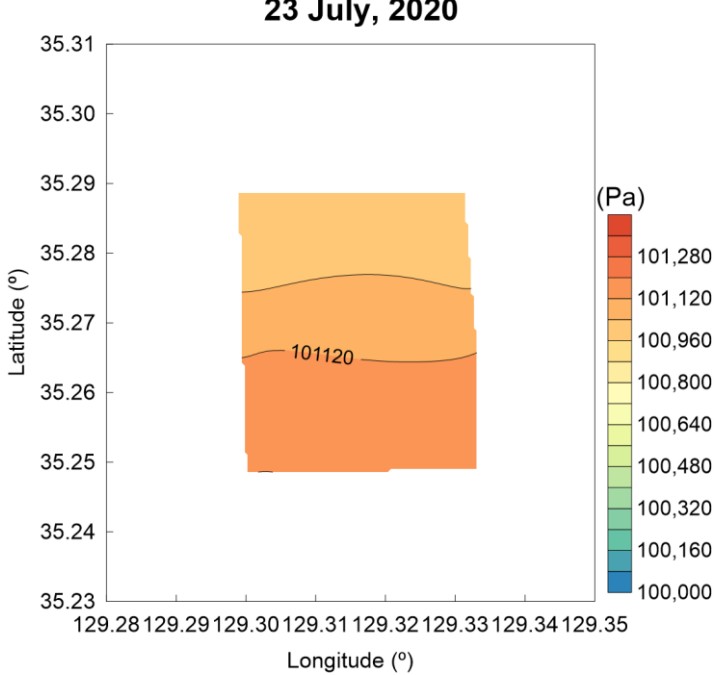

**Figure 25.** Distribution of atmospheric pressure in the Gijang coastal waters sea (23 July 2020).

## 4. Conclusions

In this study, the distribution of *N*. jellyfish by water layer and the number of *N*. jellyfish individuals appearing on the surface layer were identified using the sighting survey, and the marine environmental structure according to individual size was analyzed using the trawl survey and CTD(Conductivity Temperature Depth).

From May to July 2020, jellyfish distributed in the East China Sea and the coastal waters of Korea were calculated and compared in volume units for comparison using

different survey methods. As a result, the jellyfish population was evenly distributed in all water layers.

The ranges of seawater temperature and salinity in May 2020 were 12.6–20.3 °C and 30.5–34.1 psu, respectively. While the seawater temperature and salinity of the main swimming depths of the jellyfish ranged between 12.9–19.5 °C and 31.1–33.9 psu, respectively. In July 2020, the seawater temperature and salinity ranged between 12.5–24.9 °C and 30.6–34.7 psu, respectively, and those of the main swimming depths ranged between 14.1–24.9 °C and 30.7–34.0 psu, respectively. Considering the atmospheric pressure fluctuation, *N.* jellyfish were widely distributed in the surface layer in the low pressure with the range between 100,900 to 101,100 Pa, but there were no significant changes in the distribution by water layer.

*N.* jellyfish are not restricted by the thermocline and are distributed evenly at low seawater temperatures. Thus, it is suggested that the distribution of the jellyfish is not significantly affected by the seawater temperature [5]. However, [11] suggested that the increase in the polyp population of *N.* jellyfish is due to an increase in seawater temperature. As the seawater temperature increases due to global warming, transverse fission of *N.* jellyfish polyps is induced. When the seawater temperature decreases, the formation of transverse fission can be delayed [12,13]. Seawater temperature is a factor for the population increase of the jellyfish, but it is suggested that seawater temperature does not have a significant effect on the habitat after growth into adults. In addition, according to [14], a previous study, *N.* jellyfish were distributed at a depth of about 40 m or less in high temperature and low salinity waters in the upper water temperature and salinity, and in a wide range of salinity of 28.6–34.7 psu. Additionally, as a result of studying the behavioral patterns of *N.* jellyfish by attaching pop-up tags, it was confirmed that vertical motion was also performed at a depth of 150 m, where the water temperature was 10 °C or more lower than the surface water temperature [15].

In the East China Sea, the average distribution densities measured using the acoustic, sighting, and trawl surveys were 8355.7 ($10^{-6}$ ind/m$^3$), 162.2 ($10^{-6}$ ind/m$^3$), and 792.5 ($10^{-6}$ ind/m$^3$), respectively. The average distribution density in the coastal waters of Korea was 2238.7 ($10^{-6}$ ind/m$^3$) for the acoustic survey, 664.9 ($10^{-6}$ ind/m$^3$) for the sighting survey, and 432.9 ($10^{-6}$ ind/m$^3$) for the trawl survey. The average distribution density of the 21 July acoustic survey conducted in the coastal waters of Gijang was 1024.5 ($10^{-6}$ ind/m$^3$), and the sighting survey showed 48.8 ($10^{-6}$ ind/m$^3$). The acoustic survey conducted on 22 July 2020 was 393.3 ($10^{-6}$ ind/m$^3$) and the sighting survey resulted in 19.6 ($10^{-6}$ ind/m$^3$). The average distribution densities measured on 23 July 2020 for acoustic and sighting surveys were 99.0 ($10^{-6}$ ind/m$^3$), and 197.2 ($10^{-6}$ ind/m$^3$), respectively. The results of the acoustic, sighting and trawl surveys were compared and all surveys except for the survey conducted on 23 July 2020 showed that the acoustic survey showed the higher distribution density compared to other survey methods. The reason the distribution density of the acoustic irradiation was the highest was because the range of the depth of the irradiation was the widest. However, since the three investigation methods complement each other's inviolable areas, it is necessary to investigate them in parallel. According to the report of [16], from 2005 to 2007, the distribution density of *N.* jellyfish in the East China Sea in August was 4.2 (n/$10^{-4}$ m$^2$) in 2005 and 6.5 (n/$10^{-4}$ m$^2$) in 2006. In 2007, 7.8 (n/$10^{-4}$m$^2$) appeared in the largest quantities in 2007. Ref. [17] conducted the sighting surveys in the East China Sea and the Yellow Sea by a Japanese research team on a passenger ship in July from 2006 to 2010. The distribution density was 2.0 (n/$10^{-2}$m$^2$) in 2006, 3.2 (n/$10^{-2}$m$^2$) in 2007, and 0.02 (n/$10^{-2}$m$^2$) in 2008 [2]. Ref. [18] identified the number of surface *N.* jellyfish populations at a maximum of 120 m from Sangchuja Island in July 2020 using a drone, and as a result, a total of 173 were detected, and the distribution density was about 1.49 (n/100 m$^2$).Due to the assumption that the number of individuals increases due to global warming, the distribution density of *N.* jellyfish is expected to increase by 2020. Small individuals of 20 to 50 cm in length appearing in the East China Sea were mainly found in low salinity where the Yangtze River runoff flows towards the coastal waters of Korea. It

extends northeastwards and flows into the coastal waters of Korea and Japan [2]. The size of jellyfish collected from the coastal waters of Korea was about three times larger than the size of the jellyfish collected from the East China Sea.

In addition, the higher number of *N.* jellyfish individuals appeared in the surface layer during rainy weather as detected during the sighting survey. It is assumed that the jellyfish migrates to the surface due to the influence of atmospheric pressure and precipitation.

The results of this study can be used as basic data to predict the appearance of *N.* jellyfish according to the marine environmental conditions.

**Author Contributions:** Conceptualization, S.O. and K.-Y.K.; methodology, S.O.; software, W.O.; validation, G.P., H.-J.O. and W.O.; formal analysis, S.O.; investigation, S.O. and K.-Y.K.; resources, K.L.; data curation, G.P.; writing—original draft preparation, S.O.; writing—review and editing, G.P.; visualization, S.O.; supervision, K.L.; project administration, K.L.; funding acquisition, H.-J.O. All authors have read and agreed to the published version of the manuscript.

**Funding:** This research was funded by a grant from the National Institute of Fisheries Science, Korea (2022052) and was partially supported the project titled "Development of AI Based Smart Fisheries Management System (No. 20210499)" funded by the Ministry of Oceans and Fisheries, Korea.

**Institutional Review Board Statement:** Not applicable.

**Informed Consent Statement:** Not applicable.

**Data Availability Statement:** Not applicable.

**Acknowledgments:** This research were funded by grant from National Institute of Fisheries Science, Korea (2022052) and was partially supported the project titled "Development of AI Based Smart Fisheries Management System (No. 20210499)" funded by the Ministry of Oceans and Fisheries, Korea. We are grateful to two anonymous reviews for helpful improving this paper.

**Conflicts of Interest:** The authors declare no conflict of interest.

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
