# Peer review of "Spatio-Temporal Distribution of Giant Jellyfish (Nemopilema nomurai)"

_water, doi:10.3390/w14182883_

Round 1

Reviewer 1 Report

Comments on the manuscript “Spatio-Temporal distribution of Giant jellyfish(Nemopilema nomurai) “by Sun Young Oh et al.

This is a very interesting manuscript.  The presented work applies acoustic techniques to describe spatio-temporal distribution of Giant jellyfish (Nemopilema nomurai).  To do it, EK60 and EK80 Simrad echosounders were used working at 38 and 120 kHz. In both cases, split-beam transducers were used. Acoustic data obtained are compared with sighting and trawl surveys results. The comparison between acoustic survey and other techniques suggests that acoustic survey results are higher than  any of the other methods. Authors conclude that the jellyfish population is distributed through water column. Moreover, they observed that jellyfish were not affected by the thermocline and a jellyfish vertical migration was influenced by atmospheric pressure and precipitation.

In general, this is a very well written and well-structured manuscript. The manuscript is suitable for publication in the journal. This reviewer has a major concern on the manuscript:

As established in (Kim et al., 2020) , It is necessary to obtain TS values at 38 and 120 kHz  or SV values for both frequencies. On the basis of TS or SV results obtained, the mean volume backscattering strength (∆MVBS) at multiple frequencies is calculated. Given that the performance measurement of the method depends on the ability of the echosounder to provide accurate  measurements, a good calibration of the acoustic equipment should be made. Therefore, I would suggest adding more detail about echounder calibration, at least in the 2.2 section. In the same way parameters used to obtain individual echoes should be described in 2.3 section. I also think that the paper would benefit if SV or TS data used  will be shown using  either a graph or table display.

Minor comments

Lines 106-112: The acoustic signal identification method is insufficiently described in my opinion. An improvement of Figure 4 would be welcome here. 

Lines 349-374: In this paragraph, the discussion has centered on a comparison of acoustic survey, sighting survey and trawl survey.  The conclusion drawn  is that:  "...acoustic survey showed the higher distribution compared with other methods…' '. A more detailed comparison of the results of three methods is lacking in this part.

Would be very welcome that the minor issues exposed, will be better explained and clarified in detail to the better understanding of results depicted, and applied methodology.

Reviewer 2 Report

This manuscript provided some information of the jellyfish which occurred in Eastern China sea and Korea coastal, the survey area is wide, but the most important data present in this ms was the density, with different ways to measure, the discussion part was a kind of repeat of the results, did not show any new idea or analysis. 

The introduction part can be improved by given more information about this kind of the research, and what is the your aim

The method and matrial part, should introduce the study area, method detaily

The discussion part should add some analysis, such as distribution pattern and its driving factors, the relationship between the jellyfish and the enviromental parameters etc. 

Author Response

The author is very grateful for the content pointed out by the reviewers, and I think that the completeness of the thesis has been improved. The detailed responses of the judges are as follows.

Reviewer 3 Report

This is potentially an interesting study with a good solid survey design. However, the authors focus on giving us the absolute numbers from their surveys with (i) little context to other surveys of this species, (ii) densities of other species in other places and most significantly (iii) a detail analysis/discussion of the major discrepancies in the estimates from the different methods used. The latter would be of much wider interest to journal readers than the absolute values in this particular locality.

Make the figures bigger - easier to read !

Do not repeat data in Tables and Figures - move some material to on-line supplement if possible.

Author Response

(The authors gave the same response as above.)

Reviewer 4 Report

1)    English grammar, terminology,  and style need improvements for a scientific paper. Use of the international system of units is required

2)    That is ind abbreviation in the abstract for densities.

3)    Table 1 – change date format to conventional in the scientific literature

4)    Figures 1-3 require more explanation. What does acoustic station imply? Was the ship stationary

5)    What does the ship prefecture side mean? Do you mean the port side

6)    What is shown in Figure 4 (bad quality, no axes). I do not see any an umbrella-shade-shaped acoustic signal mentioned in Lines 106-108. Equations 1-3 use nonstandard notations and cannot be understood by a reader. Acoustic data analysis section should be fully rewritten and should include much more details. They should address how they know that the echo is returned from the jelly fish of particular kind they are interested in and not from zooplankton or fish school

7)    Figure 5 is difficult to interpret, provide the color synch for the axes corresponding to red and blue curves. Describe histograms, are they densities

8)    Lines 131 : not density units are used to describe density (ind/nmi is not a density unit). The authors never discussed how they determine the density from individual counts at the station

9)    Line 138 : what is the density of acoustic irradiation (line 143)?

10) Line 155-157: what is hPa? Use scientific notations and conventional units

11) Figure 6 and 8 : what is the color scale measures?

12) Explain how the different depth profiled and what is the depth resolution?

13) What was the detection range for the echosounder in use?

14) Very limited references on similar studies across different regions

15) The orders of magnitude differences in densities determined by different methods are not addressed: could be the mathematical errors since the details of the analysis are not discussed.

16) Lines 361-368: the results are compared to the density values measured in totally different units.

Author Response

(The authors gave the same response as above.)

Round 2

Reviewer 2 Report

The author made minor revision on the manuscript, that's better than the former version. I think it is ok to be published in this journal, because the jellyfish play important role in ecosystem, and to collect them is not an easy work.